# Obesity under full fresh fruit and vegetable access conditions

**Andres Silva**[1]*, **Pilar Jano**[2], **Nicolás Von Hausen**[3]

**1** Department of Economics, Government and Communication, Universidad Central de Chile, Santiago, Chile, **2** Business School and Center of Economics and Regional Policy (CEPR), Universidad Adolfo Ibáñez, Viña del Mar, Chile, **3** Analysis Unit, Servicio Nacional de Capacitación y Empleo, Ministerio del Trabajo, Santiago, Chile

* andres.silva@ucentral.cl

**Data Availability Statement:** The dataset is holded by the Subsecretaria de Salud Pública del Ministerio de Salud de Chile. The dataset is publicly available by request at https://www.portaltransparencia.cl/. File name: ENS-2009-2010-DEPTO.EPIDEMIOLOGIA-MINSALSTATA-Version.

## Abstract

There is no agreement regarding the role of fresh fruit and vegetables' affordability, accessibility and availability, or access in general, on obesity rates. In this article we investigated whether access to fresh fruit and vegetables is related to better biometric indicators such as weight and body mass index. Using mediation and matching methods and assuming that farmers and traditional market sellers have easy access to fruit and vegetables, we found that having better access is not associated to a reduction in weight or body mass index. Potential explanations for this result are that better access was not associated with fresh fruit and vegetables' consumption and fruit and vegetables' consumption was not associated with weight and body mass index. Also, fresh fruit and vegetables' sellers had a higher weight and body mass index compared to the rest of the population but, a similar weight and body mass index compared to people with their same educational level. Therefore, variations on weight and body mass index were more associated with educational level rather than with access. Access may not be the single story to explain fruit and vegetable consumption.

## 1 Introduction

Fresh fruit and vegetables (FVs) are a natural source of vitamins, minerals and fiber and a critical component of a balanced diet [1]. A diet high in fresh FV leads to better health outcomes [2]. However, consuming recommended amounts of fresh FV can be constrained by access in terms of affordability, accessibility and availability. Affordability happens when a household is able to purchase its desirable food basket given its price. Accessibility corresponds to how easy it is to get to a food store location. Availability is the possibility to find the product in the store. Therefore, food access, at least, is a broad term that considers affordability, accessibility and availability.

The purpose of this article is to investigate whether a segment of the population with easy access to fresh FV has better biometric indicators, such as weight and Body Mass Index (BMI), compared to the rest of the population. Using the *Encuesta Nacional de Salud* (ENS, Chilean National Health Survey), we found that having better access is not associated to a reduction in

rar. The documentation is available at http://www.
repositoriodigital.minsal.cl/handle/2015/601.

**Funding:** This work was funded by the National
Agency for Research and Development (ANID) /
FONDECYT de Iniciación/2020 – 11201115. The
funders had no role in study design, data collection
and analysis, decision to publish, or preparation of
the manuscript.

**Competing interests:** No authors have competing
interests.

weight or BMI. Potential explanations for this result are that: 1) according to a mediation analysis, better access was not associated with FV consumption and FV consumption was not associated with weight and BMI; and 2) according to a propensity score matching analysis, fresh FV sellers had a higher weight and BMI compared to the rest of the population but, a similar weight and BMI compared to people with their same educational level. Therefore, variations on weight and BMI were more associated with educational level rather than with access. For example, we found that higher education is linked with higher FV consumption and lower BMI.

Fresh FV sellers represent a population segment that can inform public policies. Based on our findings and with the objective of increasing consumption of fresh FV, public policies that improve educational level are likelier to lead to a larger impact on fresh FV consumption than policies that only facilitate fresh FV access, in terms of accessibility, affordability and availability. While fresh FV access can be a desirable condition in any food environment, we argue that, in some cases, the effect of fresh FV access can be overstated.

## 2 Background

A variety of health policies seek to encourage the consumption of fresh FV. On the demand side, health policies focus on providing information, educating children in schools, and looking for ways to reduce fresh FV prices by means of coupons or specific subsidies. On the supply side, substantial attention is paid to the existence of areas with limited access to affordable healthy food, also known as "food deserts" [3, 4]. From the access point of view, providing coupons and specific subsidies seeks to improve affordability, while increasing food distribution seeks to improve accessibility.

Previous studies have attempted to measure the role of food accessibility on health indicators such as consumption of fresh FV *per capita* and BMI, but measuring accessibility is problematic. People do not necessarily move in straight lines; therefore, distance to the closest food shop can be a misleading accessibility indicator [5]. Moreover, car availability in the household would also change, at least, accessibility conditions [6]. Recognizing that accessibility to healthy food is likely to play a relevant role in some food environments, and possibly, may interact with other determinants, it is unclear how we can argue effectiveness of a food policy without taking into account individual food accessibility conditions.

The underlying assumption is that a household located in a food desert has difficulties purchasing fresh FV. In this way, lack of fresh FV access would lead to an unhealthy food basket, and then, to a higher prevalence of malnutrition related diseases [7]. Previous studies document the existence of areas with limited access to healthy food in the United States, the United Kingdom, and Canada [3]. Therefore, there is an agreement that not all the population has guaranteed access to healthy food.

While there is an agreement regarding the existence of food deserts in some cities, it seems there is no such agreement on the actual effect of a food desert on purchasing behaviors and diet-related diseases [8]. Some research finds that food deserts are determinants in terms of prevalence of obesity and diet-related diseases [9, 10], other research finds an ambiguous relationship [4, 11–13]. Ver Ploeg and Wilde find that households in the same neighborhood, and in the same food environment, can have different food purchasing patterns [14].

Some possible explanations of the unclear behavioral effect of food deserts may be linked with the empirical variable specification. LeClair and Aksan argue that to truly define a food desert, one would need to take into account the price-distance cost of food to understand consumer behavior, which would imply redefining food maps [15]. Furthermore, higher shopping frequency leads to less healthy food purchases, since consumers buy more temptation foods

[16]. Supermarkets, having unhealthy and healthy foods, make all food more available to households [17]. Therefore, there is a lack of agreement regarding the effect of food deserts on fresh FV consumption.

Socio-demographic and lifestyle characteristics also play a relevant role in understanding fresh FV disparities [18]. High income and highly educated households purchase higher-quality food baskets (whole grain, lean meats, fish, low-fat dairy products, fresh FV) [4, 19]. Previous research does not show the causality path [19]. The mechanism how socio-demographic characteristics lead to a food consumption pattern is unclear. It is straightforward to believe that low-income households tend to buy highly-dense food since they contain cheaper calories than healthy food; therefore, it does not make sense to promote high-cost food to low-income households [19]. However, this mechanism can oversimplify the true problem.

In this study, we build upon previous research by providing empirical evidence regarding the link between education, household income, and food access, and biometric indicators (weight and BMI). In this way, we want to inform the debate on food deserts. We analyzed fresh FV sellers since we argue that they are a particular population group. In general, fresh FV sellers are economically modest and have a low educational level. To some extent, fresh FV sellers may resemble many low-skilled workers, but with access to fresh FV while at their work places. Based on the analysis of biometric indicators of fresh FV sellers, we obtained results that we expect contribute to generating an effective strategy to increase the consumption of fresh FV. To date, to the best of our knowledge, there are no studies available that analyze biometric or health indicators of fresh FV sellers.

## 3 Materials and methods

We compared a specific population segment, fresh FV sellers, with the rest of the country's population. With this purpose, we divided our analysis into three parts. First, we compared a series of descriptive statistics (e.g. socio-demographics, disease prevalence, and biometric indicators) between these two groups. Second, using a mediation model, we explained the variability of obesity-related biometric indicators (weight and BMI) as a function of occupation (fresh FV sellers or not) while controlling for confounding factors (including income and education). Full and partial effects discussed in the original article by Baron and Kenny [20] can be better explained as direct and indirect effects [21]. Fresh FV sellers' access (independent variable) can directly affect BMI (dependent variable) or, indirectly affect BMI through higher FV consumption (mediator variable). In our analysis, we estimate a separate mediation model for weight and BMI. Finally, using alternative matching specifications, we compared fresh FV sellers with the rest of the country. These three analyses are complementary in explaining the similarities and differences of fresh FV sellers with respect to the rest of the country.

### 3.1 Data

The ENS is the largest survey in Chile that collects information on transmissible and non-communicable diseases and their main risk factors. The first survey was conducted in 2003, and the second version, in 2009-10. We used this latter version for our analyses. The ENS sample design was built based on the National Population Census to have a representative sample at the country, regional, and zone (rural/urban) levels. The survey was conducted to individuals over 15 years old. The response rate was 85%. Some respondents were contacted by phone later to complete some missing values. The complete documentation and data set are available at the Ministry of Health in Chile or from the authors upon request.

The survey included direct self-reported questions about the health status of the respondents as well as bio-physiological and biochemical measurements that involved taking samples

for laboratory tests and a set of biometric indicators. The self-reported questions included education level, household income level, marital status, age, lifestyle variables such as physical activity, fresh FV consumption, and occupation, including farmers and traditional fresh FV sellers, among other variables of interest. The biometric measures, such as waist diameter, height and weight, therefore BMI, were conducted by health-care professionals. In fact, taking into account the survey's complexity, the data collection process included the participation of nurses, who examined the respondents and sent urine and blood samples to be analyzed in laboratory facilities. The analysis of the ENS serves as an input for national health planning and for the estimation of disease burden and attributable load in Chile.

Based on these data, we generated evidence regarding the similarities and differences in biometric indicators and fresh FV consumption of fresh FV sellers and the rest of the adult population in Chile. We assumed farmers and traditional market fresh FV sellers in the sample had relatively easy access to fresh FV because, in general, farmers and traditional market fresh FV sellers are also owners. In fact, 77% of traditional market fresh FV sellers own the fresh FVs they sell [22]. In what follows, we explain the estimation methods used.

## 3.2 Mediation model

The mediation model was developed by Baron and Kenny [20]. A discussion can be found in the work presented by Iacobucci *et al.* [23] and Zhao *et al.* [21], and a review of health-related applications can be found in the work by Liu *et al.* [24]. In our case, fresh FV sellers have better access to fresh FV products, which may be linked to their FV consumption, and therefore, their weight and BMI.

Using a mediation model, we estimated these two effects in two different equations. In the first equation, we tested whether being a fresh FV seller is associated with higher FV consumption. Then, in the second equation, we tested whether FV consumption was associated to a change on weight and BMI. In our case, the indirect effect corresponds to the product of these coefficients of the mediation path. In both equations, we controlled for age, gender, marital status, education, and income. We also controlled for height in the weight model.

## 3.3 Matching model

We needed to test whether the differences in weight (in kilograms) and BMI remained after controlling for confounding factors. Given that being a fresh FV seller is expected to be non-random, we implemented a propensity score matching technique to estimate the likelihood of being a fresh FV seller based on the variables available in the survey. In this way, we needed to create a balanced sample for both groups (fresh FV sellers and the rest of the population). Mathematically, we estimated the following expression:

$$p(X_i) = Pr(D = 1|X_i) = E(D|X_i) \qquad (1)$$

Where $D$ represents the fresh FV seller occupation dummy variable, and $X$ is a group of socioeconomic variables of the fresh FV seller. For the propensity score matching to be a valid technique for estimating the likelihood of participation, three conditions must be satisfied. First, overlap, a valid counterfactual must exist, that we show graphically. Second, the sample should be balanced across the mentioned variables, given the propensity score. Third, unconfoundedness of participation given the propensity score, meaning that once the sample is balanced, the expected probability of being a fresh FV seller for all people should be the same.

Given the desire to explore the differences in biometric indicators between fresh FV sellers and similar people in the sample, we computed the average difference of the objective biometric indicators between matched people for the mentioned outcomes as the Average Treatment

Effect (ATE). Thus, the fresh FV seller differential is computed as follows:

$$\tau = E(Y_{1i}|D_i = 1, p(X_i)) - E(Y_{0i}|D_i = 0, p(X_i)) \tag{2}$$

Where $Y_1$ and $Y_0$ are the potential outcomes for obesity-related biometric variables (weight and BMI) in fresh FV sellers and the rest of the population, respectively. There are many possible matching criteria. In this article, we implemented nearest neighbor matching using three and six closest neighbors and the bilevel optimization estimator (BLOP) for robustness purposes.

In the nearest neighbor matching, each observation in the treated group is matched with the nearest N observations in the control group. The distances are computed as the score differences. However, there are many matching criteria and the value of N needs to be specified by the researcher. To the best of our knowledge, BLOP, recently developed by Díaz *et al*, is the only matching criterion that is data-driven [25]. In the BLOP estimator, each treated observation is matched with a weighted average of the control group. There are two optimization problems, finding a weight that minimizes the distance between treated and control observations and finding a weight that minimizes the distance between a treated observation and a weighted average of control observations. As a result, the two optimization problems produce a unique vector of weights for each treated observation [25].

In the coming section, we present the results, starting with the prevalence of diseases and risk factors, as well as weight categories, for fresh FV sellers compared to the rest of the national population, followed by the results of the Mediation and Matching models.

## 4 Results

### 4.1 Descriptive statistics

The upper part of Table 1 shows the prevalence of selected related-health variables by occupation (fresh FV sellers versus the rest of the country statistics). Compared with the rest of the country, fresh FV sellers present less prevalence of depression, less high back pain and less kidney damage. The middle part of Table 1 shows that fresh FV sellers practice significantly more physical activity. To some extent, it is expected that fresh FV sellers practice more physical activity since their occupation requires constant motion. Although it is possible that physical activity can explain at least part of the difference in disease prevalence, with this preliminary descriptive analysis, it is risky to infer a causal relationship.

The lower part of Table 1 shows that the prevalence of normal BMI is lower in fresh FV sellers compared with the rest of the country. However, fresh FV sellers do not have significantly higher overweight or obesity compared to the rest of the population.

In order to get acquainted with the data, Table 2 shows the descriptive statistics of biometric indicators. The first two variables are dependent variables: weight and BMI. Following that, we present confounding factors such as education and income level, as well as other controls such as gender, marital status, geographic area, zone, habits, and age. As compared with the rest of the population, fresh FV sellers have a higher weight and BMI, and are slightly taller. Most of them are male, have lower education, lower income, belong to rural population segments and consume similar amounts of fresh FV, while having a higher salt intake, compared to the rest of the population. Therefore, these descriptive statistics show relevant sociodemographic differences between fresh FV sellers and the rest of the population. FV consumption, captured as a self-reported measure, is not significantly different between subsamples, and is near 3-3.2 portions per person a day, on average. As reference, the WHO recommends, at least, five portions per person a day. A more detailed analysis will allow us to establish if these differences change after controlling for confounding factors. Our study allows us to test whether the

**Table 1. Prevalence of diseases by occupation.**

|  | Fresh FV Sellers | Rest of Country | t-stat | significance |
|---|---|---|---|---|
| diabetes | 7.00% | 7.81% | -0.24 |  |
| hypertension | 9.78% | 8.32% | 0.32 |  |
| depression symptom | 22.54% | 37.71% | -2.26 | ** |
| liver damage | 0.95% | 2.70% | -1.11 |  |
| lower back pain | 19.93% | 17.93% | 0.33 |  |
| higher back pain | 4.36% | 10.45% | -1.68 | * |
| kidney damage | 0.55% | 1.61% | -1.79 | * |
| cardiac risk high or very high | 11.24% | 14.46% | -0.96 |  |
| sleeping disorder | 65.90% | 63.54% | 0.31 |  |
| physical activity less than 1 hr/week | 10.13% | 36.51% | -7.22 | *** |
| physical activity more than 7 hrs/week | 45.77% | 23.51% | 3.00 | *** |
| normal | 16.02% | 34.22% | -3.37 | *** |
| overweight | 46.68% | 39.08% | 1.04 |  |
| obesity | 33.08% | 22.72% | 1.41 |  |
| morbid obesity | 4.22% | 2.22% | 0.83 |  |

Source: ENS, 2010.

*** $p < 0.01$,

** $p < 0.05$,

* $p < 0.1$.

Calculated using probability weights. The difference is tested using a regression in which the independent variable is a dummy variable that takes the value of one for a fresh FV seller and zero otherwise.

apparent greater BMI found in fresh FV sellers is associated with the occupation itself, or else, with other sociodemographic factors.

## 4.2 Mediation and matching

The results we present in this subsection indicate that having better access to FVs is not associated to a reduction in weight or BMI. Potential explanations based on the results below are that: 1) according to the mediation analysis, better access was not associated with fresh FV consumption and FV consumption was not associated with weight and BMI; and 2) according to the propensity score matching analyses, fresh FV sellers had a higher weight and BMI compared to the rest of the population but, a similar weight and BMI compared to people with their same educational level. In what follows, we present the mediation model results followed by the matching results.

Table 3 shows the mediation regression results using weight (columns 2 and 3) and BMI (columns 4 and 5) as dependent variables (outcomes). The purpose of the mediation model is to test whether being a fresh FV seller, through the mediation path, is associated to changes in the outcomes, weight and BMI. The mediation path is decomposed into two portions and each one is represented by an equation. The first mediation equation, in columns 2 and 4, tests whether being a fresh FV seller is linked with changes on FV consumption. The second mediation equation, in columns 3 and 5, tests whether FV consumption is linked with changes in the outcome variables (weight and BMI).

In the first mediation equation (columns 2 and 4), FV portions (consumption) is explained by fresh FV seller (access) and other controls such as age, gender, marital status, education, and income. In the second mediation equation (columns 3 and 5), the outcome is explained by

**Table 2. Selected variables by occupation.**

| | Fresh FV Sellers | | Rest of Country | | Difference | |
|---|---|---|---|---|---|---|
| | **mean** | **SD** | **mean** | **SD** | **t-stat** | **significance** |
| **Dependent variables** | | | | | | |
| body mass index, BMI weight, in kilograms | 28.73 | (4.67) | 27.33 | (5.72) | 2.18 | ** |
| | 78.49 | (14.89) | 72.15 | (14.63) | 2.96 | *** |
| **Gender** | | | | | | |
| female | 0.30 | (0.46) | 0.52 | (0.50) | -3.39 | *** |
| male | 0.70 | (0.46) | 0.48 | (0.50) | 3.39 | *** |
| **Marital status** | | | | | | |
| single, never married | 0.27 | (0.44) | 0.33 | (0.47) | -0.90 | |
| married couple | 0.63 | (0.48) | 0.54 | (0.50) | 1.27 | |
| divorced | 0.10 | (0.30) | 0.13 | (0.33) | -0.75 | |
| **Education level** | | | | | | |
| years of education | 8.54 | (3.83) | 10.68 | (4.01) | -3.92 | *** |
| years of education, mother | 5.92 | (3.72) | 8.04 | (4.47) | -3.42 | *** |
| **Household income level** | | | | | | |
| low, up to Ch$250 thousands | 0.68 | (0.47) | 0.49 | (0.50) | 2.40 | *** |
| mid-low, Ch$251 to 351 thousands | 0.23 | (0.42) | 0.27 | (0.44) | -0.64 | |
| mid-high, Ch$451 to 851 thousands | 0.10 | (0.30) | 0.16 | (0.36) | -1.17 | |
| high, Ch$851 thousands and more | 0.00 | (0.00) | 0.08 | (0.27) | -10.69 | *** |
| **Geographic area** | | | | | | |
| north | 0.11 | (0.31) | 0.11 | (0.32) | -0.12 | |
| central | 0.46 | (0.50) | 0.61 | (0.49) | -1.75 | * |
| south | 0.43 | (0.50) | 0.28 | (0.45) | 1.86 | * |
| **Zone** | | | | | | |
| urban_person | 0.54 | (0.50) | 0.88 | (0.33) | -4.61 | *** |
| rural_person | 0.46 | (0.50) | 0.12 | (0.33) | 4.61 | *** |
| **Habits** | | | | | | |
| number of hours of sleep | 7.25 | (1.39) | 7.49 | (1.69) | -1.23 | |
| FV portions, 1 portion = 80 g | 3.35 | (3.10) | 3.08 | (3.03) | 0.53 | |
| salt consumption g/day | 10.80 | (2.38) | 9.84 | (2.96) | 2.06 | ** |
| **Other** | | | | | | |
| age in years | 46.35 | (12.13) | 41.46 | (17.72) | 3.06 | *** |
| height, in meters | 1.65 | (0.09) | 1.63 | (0.10) | 1.86 | * |
| Observations | 96 | | 4,684 | | | |

Note: Robust standard errors in parentheses,

*** p<0.01,

** p<0.05,

* p<0.1.

A fruit and vegetable portion corresponds to 80 grams. Calculated using probability weights. The difference is tested using a regression in which the independent variable is a dummy variable that takes the value of one for a fresh FV seller and zero otherwise.

FV portions, fresh FV seller, and the same controls included in the first mediation equation. The equation that estimates weight as outcome, also includes height as a control variable.

The mediation equation results indicate that being a FV seller is not associated with consuming more FVs (columns 2 and 4) and FV consumption is not associated with weight or BMI (columns 3 and 5). For the weight equation, the indirect effect is 0.012, while for the BMI

**Table 3. Mediation regressions of BMI and weight.**

| Variables | Weight Model | | BMI Model | |
|---|---|---|---|---|
| | FV (portions) | Weight (kgs) | FV (portions) | BMI |
| fv_portions | | 0.0291 | | -0.00115 |
| | | (0.0726) | | (0.0293) |
| fresh FV seller | 0.397 | 5.592* | 0.397 | 1.890* |
| | (0.604) | (2.963) | (0.604) | (1.199) |
| age (years) | 0.0121*** | 0.0900*** | 0.0121*** | 0.0385*** |
| | (0.00280) | (0.0142) | (0.00280) | (0.00558) |
| female | 0.242*** | 0.152 | 0.242*** | 0.831*** |
| | (0.0804) | (0.550) | (0.0804) | (0.160) |
| married | -0.0382 | 5.137*** | -0.0382 | 1.959*** |
| | (0.102) | (0.500) | (0.102) | (0.202) |
| divorced | -0.0490 | 2.142*** | -0.0490 | 0.846*** |
| | (0.139) | (0.683) | (0.139) | (0.276) |
| mid education (8– 12 years) | 0.372*** | -0.157 | 0.372*** | -0.306 |
| | (0.108) | (0.533) | (0.108) | (0.214) |
| high education (more than 12 years) | 0.611*** | -0.957 | 0.611*** | -0.715** |
| | (0.147) | (0.733) | (0.147) | (0.293) |
| mid-low income | 0.230** | -0.424 | 0.230** | -0.297 |
| | (0.0976) | (0.479) | (0.0976) | (0.194) |
| mid-high income | 0.346*** | 1.412** | 0.346*** | 0.438* |
| | (0.128) | (0.629) | (0.128) | (0.255) |
| high income | 0.309* | -0.842 | 0.309* | -0.538 |
| | (0.180) | (0.886) | (0.180) | (0.357) |
| height | | 74.90*** | | |
| | | (2.986) | | |
| constant | 1.745*** | -55.84*** | 1.745*** | 24.69*** |
| | (0.181) | (5.147) | (0.181) | (0.364) |
| var(e.fv_portions) | 6.884*** | (0.144) | 6.884*** | (0.144) |
| var(e.bmi) | 165.6*** | (3.458) | 27.14*** | (0.567) |
| Observations | 4,588 | 4,588 | 4,588 | 4,588 |

Note: Robust standard errors in parentheses,

*** p<0.01,

** p<0.05,

* p<0.1.

Columns 2 and 3 show the mediation regression results using FV consumption as a mediator and weight as a dependent variable. Columns 4 and 5 show the mediation regression results using FV consumption as a mediator and BMI as a dependent variable. Fresh FV means fresh fruit and vegetables and FV means FVs.

equation, it is -0.0004. These magnitudes were calculated multiplying the FV seller coefficient by the FV portions coefficient, for each of the outcome variables. Table 4 shows that none of these indirect effects are significantly different from zero. Therefore, consistently, there is no evidence that FV consumption is acting as a mediator. According to the Baron and Kenny approach, in our results there is direct-only nonmediation (no mediation).

As mentioned, the main result from the regression on FV portions is that being a FV seller is not associated with a change on FV consumption (see Table 3 columns 2 and 4). Additional results obtained from these regressions on FV portions indicate that being a woman is associated with a higher FV consumption than being a man, and they consume, on average, 0.2 FV

**Table 4. Significance testing of indirect effect on biometric indicators.**

| Estimates | Weight | | BMI | |
|---|---|---|---|---|
| | Sobel | Monte Carlo | Sobel | Monte Carlo |
| Indirect Effect | 0.012 | 0.011 | -0.000 | -0.000 |
| | (0.034) | (0.056) | (0.012) | (0.021) |
| z-value | 0.342 | 0.206 | -0.039 | -0.016 |
| p-value | 0.732 | 0.837 | 0.969 | 0.987 |

Note: Standard errors in parentheses. The standard errors for the Monte Carlo approach are obtained after bootstrapping the same number of repetitions than the observations. These results are obtained using the MEDSEM command in Stata developed by Mehmetoglu [43]. The two test results show no evidence of mediation effect on weight or BMI.

portions more a day. Having a high education level is associated to 0.6 portions more compared to having a low education level, and having a high income level is associated to 0.3 FV portions more a day compared to a low-income level. These two coefficients, the coefficient of high income and of high education on FV consumption, are not significantly different between each other.

The main result obtained from the regressions on biometric indicators presented in Table 3 (columns 3 and 5) is that FV consumption is not associated with weight or BMI. Some additional results from these regressions are that age, gender (female), not being single, and having a middle income level compared to a low income level are associated to higher BMIs. Moreover, a high education is associated to lower BMIs, compared to a low education. These regressions also show that being a FV seller is associated to higher weight and BMI compared to the rest of the population. In fact, FV sellers' weight is 5.6 kilograms higher, while their BMI is 1.9 units higher. Therefore, despite the easy fresh FV access, FV sellers have a significantly higher weight and BMI compared to the rest of the population.

Finally, we conducted a set of propensity score matching. Our treated group consisted of individuals that were fresh FV sellers, while our control group was formed by the remaining subjects in the database. For clarification, we have not conducted an experiment. Assigning a group as treatment and a group as control is part of the matching procedure. Table 5 shows

**Table 5. Average treatment effect using alternative specifications.**

| Variables | Full Sample | | Low Education Sample | |
|---|---|---|---|---|
| | Weight (kgs) | BMI | Weight (kgs) | BMI |
| 3 nearest-neighbor | 9.134*** | 2.796*** | 6.770 | 2.644 |
| | (2.152) | (0.749) | (4.613) | (1.865) |
| 6 nearest-neighbor | 7.978*** | 2.541*** | 4.291 | 1.602 |
| | (2.019) | (0.578) | (3.403) | (1.458) |
| BLOP matching | 6.238*** | 1.827*** | 4.334 | 0.972 |
| | (1.956) | (0.758) | (2.823) | (0.740) |

Note: Robust standard errors in parentheses,

*** $p < 0.01$,

** $p < 0.05$,

* $p < 0.1$.

Columns 2 and 3 show the Average Treatment Effect (ATE) using the full sample. Columns 4 and 5 show ATE using only the low education subsample.

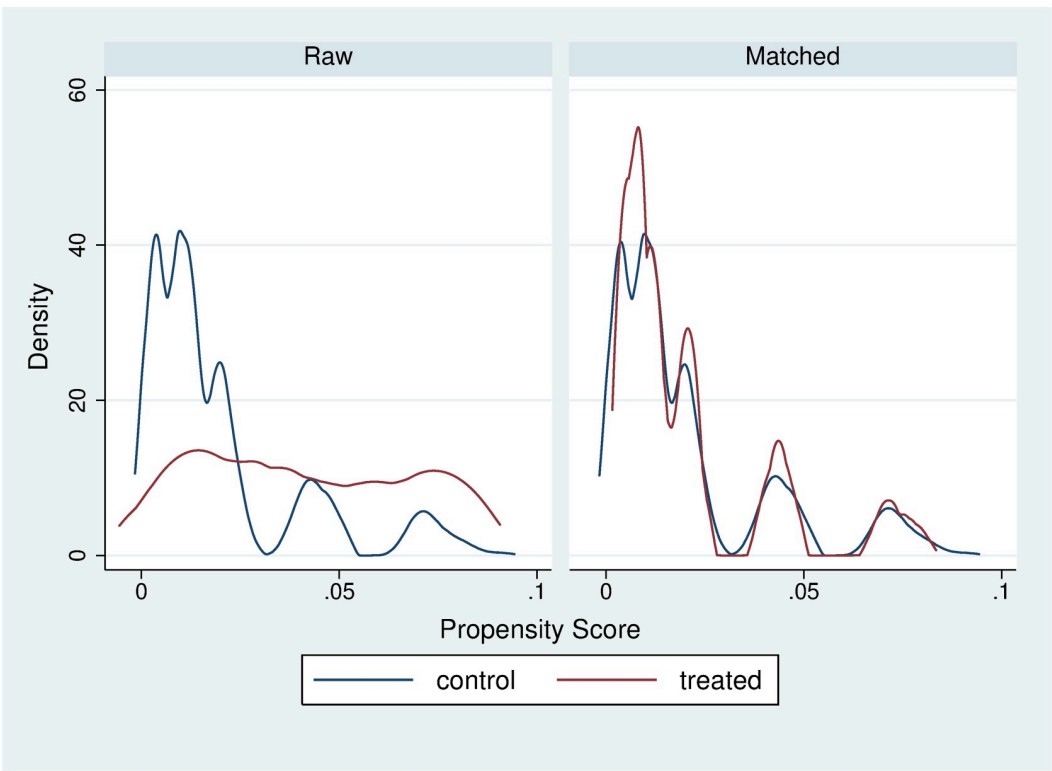

**Fig 1. 6-nearest neighbor weight matching balance.**

three alternative specifications using the full sample (columns 2 and 3) and a subsample considering only the population with low education level (less than eight years of formal education, columns 4 and 5). Using the full sample, the three matching specifications consistently show that fresh FV sellers weight around 6 to 9 kilograms more than the control group and have a BMI 2 to 3 units higher. Using the low educated subsample, there is not a significant difference between fresh FV sellers and the rest of the population. According to the matching results, the variations on weight and BMI seem more linked with differences in education level rather than with being a FV seller.

One of the assumptions required to use the matching estimators is the overlap assumption, which states that each individual has a positive probability of receiving each treatment level. The overlap assumption is satisfied when there is a chance of seeing observations in both the control and the treatment groups at each combination of covariate values. Figs 1 and 2 show the plots of the estimated densities of the probability of being a fresh FV seller. These plots can be used to check whether the overlap assumption is violated. Finally, Tables 6 and 7 present the covariate balance summary for outcome variables (weight and BMI).

Comparing the results obtained through mediation and matching, we observe that fresh FV sellers have higher obesity indicators than the rest of the population. We consistently found, throughout our estimations, that having better fresh FV access is not associated to a lower BMI. Our results suggest that fresh FV sellers' BMI is more linked to their education level rather than to their fresh FV access conditions.

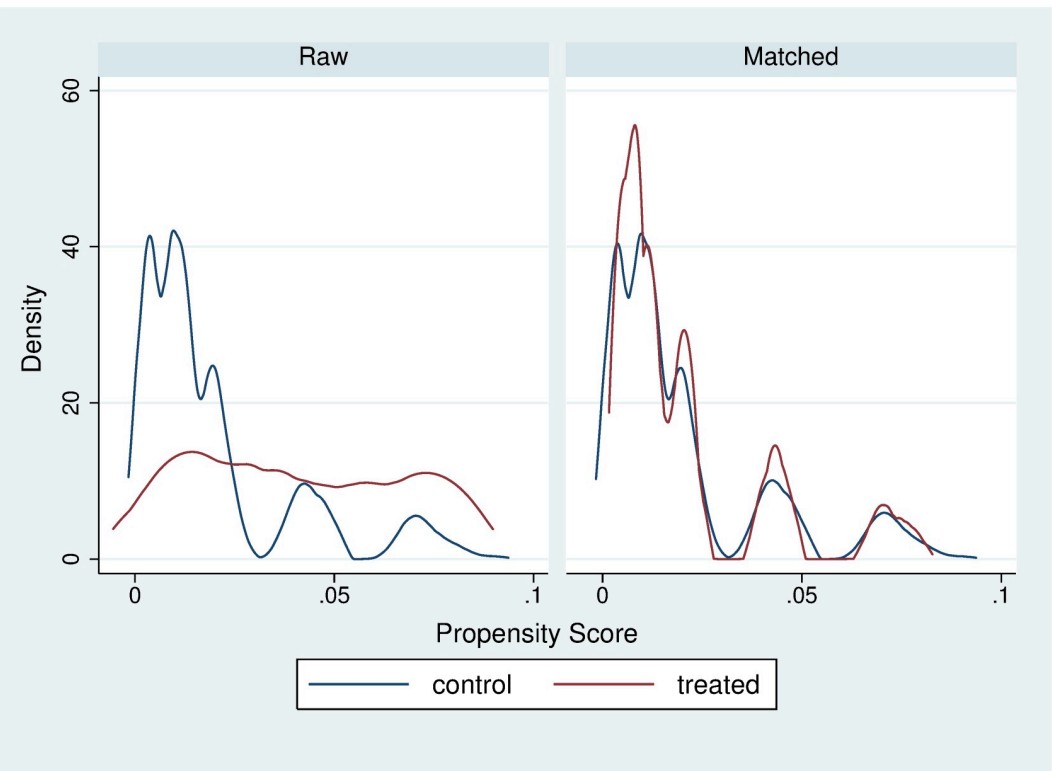

**Fig 2. 6-nearest neighbor BMI matching balance.**

## 5 Discussion

Regarding the debate on ways to increase consumption of fresh FV, previous literature has mainly pointed at affordability, accessibility, and availability. As a part of accessibility, food deserts, that is, places with limited access to healthy food, have been suggested as determinants of obesity [3, 4]. As of affordability, a healthy food basket tends to be more expensive than an unhealthy one in terms of energy density [26, 27]. In fact, there is an inverse relation between energy density (megajoule/kilogram) and energy cost (US$/megajoule), such that energy-

**Table 6. Covariate balance summary weight matching 6 nearest-neighbor.**

|  | Raw | Matched |  |  |
|---|---|---|---|---|
| Number of observations | 4,625 | 9,250 |  |  |
| Treated observations | 92 | 4,625 |  |  |
| Control observations | 4,533 | 4,625 |  |  |
|  | Standardized differences |  | Variance ratio |  |
|  | Raw | Matched | Raw | Matched |
| age | 0.216 | 0.001 | 0.516 | 0.493 |
| woman | -0.657 | -0.014 | 0.877 | 1.005 |
| marital status—single | 0.043 | 0.103 | 0.998 | 0.961 |
| marital status—married | 0.040 | 0.216 | 0.946 | 0.648 |
| education—mid | -0.120 | 0.044 | 1.020 | 0.990 |
| education—high | -0.510 | -0.077 | 0.210 | 0.873 |
| income | -0.589 | -0.265 | 0.355 | 0.510 |

**Table 7. Covariate balance summary BMI matching 6 nearest-neighbor.**

|  | Raw | Matched |  |  |
|---|---|---|---|---|
| Number of observations | 4,596 | 9,192 |  |  |
| Treated observations | 92 | 4,596 |  |  |
| Control observations | 4,504 | 4,596 |  |  |
|  | Standardized differences | | Variance ratio | |
|  | Raw | Matched | Raw | Matched |
| age | 0.218 | 0.007 | 0.519 | 0.498 |
| woman | -0.657 | -0.012 | 0.877 | 1.005 |
| marital status—single | 0.042 | 0.094 | 0.998 | 0.965 |
| marital status—married | -0.037 | -0.213 | 0.950 | 0.651 |
| education—mid | -0.121 | 0.045 | 1.020 | 0.989 |
| education—high | -0.513 | -0.078 | 0.208 | 0.871 |
| income | -0.592 | -0.268 | 0.354 | 0.508 |

dense foods composed of refined grains, added sugars, or fats may represent the lowest-cost option to the consumer [26]. Also, subsidies implementation, vouchers distribution, and Value Added Tax (VAT) decreases have been studied as alternatives to reduce fresh FV prices, making them more affordable [28, 29]. The underlying assumption is that making fresh FV more accessible and more affordable, would improve consumers' intake of fresh FV.

In this article, we revisited fresh FV access as an obesity determinant to inform the debate. We used the ENS in Chile to assess fresh FV sellers' obesity-related indicators such as weight and BMI levels. This population segment has guaranteed access to fresh FV, allowing us to control for fresh FV access in our estimations.

In order to investigate if better FV access is related to a reduction in obesity indicators, we used two different methodologies: mediation regression and propensity score matching. We found that having better fresh FV access is not related to a reduction in weight or BMI. Our mediation analyses results show that better access was not associated with FV consumption and FV consumption was not associated with weight or BMI. In other words, we found no evidence that FV consumption acts as a mediator in the regression of FV seller (representing access) on weight or BMI. Our propensity score matching analyses indicate that fresh FV sellers had a higher weight and BMI compared to the rest of the population but, a similar weight and BMI compared to people with their same educational level. So, variations on weight and BMI were more associated with educational level rather than with access.

Based on our results, public policies that focus on improving fresh FV access might not lead to changes in BMI. Our findings are consistent with some of the previous research. For example, some studies on individual grocery store entry show that improving neighborhood FV access did not change residents' FV consumption or BMI [30, 31].

As to the relation between FV consumption and weight/BMI, the evidence in the literature is mixed and seems to depend upon methodology. For example, randomized controlled trials (RCTs) show that an increase in FV consumption has either no effect on body weight, or a small reduction [32, 33]. Prospective observational studies, in general, show that consuming more FVs reduces anthropometric parameters and risk of overweight/obesity, abdominal obesity, or weight gain [34, 35]. Therefore, our result that relates FV consumption with weight/BMI is more consistent with results obtained in RCTs. Our result is also consistent with another study conducted in South America (Perú), that finds that greater FV consumption is unrelated to overweight or obesity [36]. Consistent with our results, the later study also finds that overweight or obesity is related to education level and socioeconomic status. Our result

that high income is related to higher FV consumption compared to low income is also consistent with Middaugh and coauthors' finding that lower-income households in the US, consume less fresh FV than higher-income households [37].

We also found that individual characteristics such as gender, marital status, and age are also significantly correlated with BMI. Therefore, if programs aiming to reduce BMI want to be more effective, they should also consider the profile or characteristics of people that tend to have higher BMI, who in our results tend to be women, married individuals, divorced individuals, and older people.

We cannot extrapolate our findings to any food environment, especially considering Hawkes' [38] recent argument that food availability has a multi-dimensional nature that cannot be explained by any single story. However, our results may potentially be extrapolated to food environments where access to fresh FV is either granted or cheap. Further, we argue that our results may help rethink some food policies. Price changes, such as dropping the VAT from fresh FV, represent a fiscal cost while their actual effect on BMI is unclear [39, 40]. For example, experimental evidence shows that price discounts in healthier foods make more impulsive individuals increase their purchase of less healthy, high-energy-dense foods [41]. Our findings suggest that creating incentives to locate food stores, giving fresh FV vouchers, and donating fresh FV baskets may not lead to significant BMI reductions if these policies are not associated with an education campaign.

Educational strategies, rather than eliminating food deserts, are likelier to be effective in improving diet quality and reducing food inequalities [4, 42]. Consistently, we found that high education level was associated to a lower BMI and a higher FV consumption. Of course, we cannot argue that food deserts would not lead to any impact in any circumstance. However, evidence in this article shows that education is more linked to FV consumption than fresh FV access. Therefore, education should be considered when designing obesity-related policies and programs.

Our study is not free from limitations. The cross-sectional nature of our data as well as having a relatively small sample of fresh FV sellers, may be a limitation. In addition, the survey was not designed with the intention of measuring the fresh FV sellers' lifestyle. Finally, another potential limitation is that our argument that fresh FV sellers have better access to fresh FV products, cannot be tested directly.

In the future, we would be interested in deepening our understanding of the relationship among income, education, fresh FV access, and health. We also wish to conduct experimental work to change consumption behavior. For example, we are interested in investigating how different treatments in a target population can improve consumption of fresh FVs.

## Acknowledgments

We wish to thank Gloria Tarres for improving the English, grammar and flow of the article. Any errors and shortcomings are our own. The views expressed in this article are those of the authors and do not necessarily represent those of their institutions. Ethical approval was not required for this work as no new empirical data were collected.

## Author Contributions

**Conceptualization:** Andres Silva, Pilar Jano, Nicolás Von Hausen.

**Data curation:** Andres Silva, Nicolás Von Hausen.

**Formal analysis:** Andres Silva, Pilar Jano, Nicolás Von Hausen.

**Investigation:** Pilar Jano.

**Methodology:** Andres Silva.

**Validation:** Andres Silva.

**Visualization:** Andres Silva.

**Writing – original draft:** Andres Silva.

**Writing – review & editing:** Andres Silva, Pilar Jano.

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
