## [Decision Letter · Decision Letter 0]

20 Apr 2020

PONE-D-20-02163

Obesity under Full Fruit and Vegetable Access Conditions: Expected and Unexpected Results

PLOS ONE

Dear Dr. Silva,

Thank you for submitting your manuscript to PLOS ONE. After careful consideration, we feel that it has merit but does not fully meet PLOS ONE’s publication criteria as it currently stands. Therefore, we invite you to submit a revised version of the manuscript that addresses the points raised during the review process.

The revised version should address all comments in the reports.

We would appreciate receiving your revised manuscript by Jun 04 2020 11:59PM. To enhance the reproducibility of your results, we recommend that if applicable you deposit your laboratory protocols in protocols.io, where a protocol can be assigned its own identifier (DOI) such that it can be cited independently in the future. For instructions see: http://journals.plos.org/plosone/s/submission-guidelines#loc-laboratory-protocols

We look forward to receiving your revised manuscript.

Kind regards,

Petri Böckerman

Academic Editor

PLOS ONE

Journal Requirements:

2. Please expand the abstract of your study, following PLOS ONE submission criteria (https://journals.plos.org/plosone/s/submission-guidelines#loc-abstract).

3. Please include a caption for figures 1 and 2.

4. Please remove your figures from within your manuscript file, leaving only the individual TIFF/EPS image files, uploaded separately.  These will be automatically included in the reviewers’ PDF.

<h3>** **</h3>

5. Please ensure that you refer to Figure 2 in your text as, if accepted, production will need this reference to link the reader to the figure.

6. We note you have included tables to which you do not refer in the text of your manuscript. Please ensure that you refer to Tables 5-11 in your text; if accepted, production will need this reference to link the reader to each Table.

Reviewers' comments:

Reviewer's Responses to Questions

**Comments to the Author**

1. Is the manuscript technically sound, and do the data support the conclusions?

Reviewer #1: No

Reviewer #2: Yes

2. Has the statistical analysis been performed appropriately and rigorously? 

Reviewer #1: No

Reviewer #2: Yes

3. Have the authors made all data underlying the findings in their manuscript fully available?

Reviewer #1: No

Reviewer #2: Yes

4. Is the manuscript presented in an intelligible fashion and written in standard English?

Reviewer #1: No

Reviewer #2: Yes

5. Review Comments to the Author

Reviewer #1: GENERAL

This paper is purportedly about fruit and vegetable “access,” but concepts are poorly defined, and often conflated. The authors makes unfounded assumptions and arrive at unsupported conclusions. There is lack of recognition about the substantial limitations in the study design. The text would benefit from review by someone who is a native English speaker, although problems with English are not the main issues.

SPECIFIC

Overall:

- Constructions like “According to [4]” (line 10) are quite odd. Other examples of similar constructions appear on all of the following lines: 65, 71, 74, 140, and 274.

Abstract:

- “access” meaning what? (even parenthetically) One or more of five dimensions discussed later?

- “education” defined how? (even parenthetically) Years of schooling? Subject-matter expertise?

- “income” defined how? (even parenthetically) For individuals / households? Absolute / relative?

- What are “produce sellers”? Farm-stand workers? Farmers? Produce-store owners? Fruit-cart vendors?

- What does “granted access” mean?

Intro:

- Line 2: Fruits and vegetables do not have to be “fresh” to be “produce.” Frozen, canned, dried, and jarred fruits and vegetables also count.

- Line 3: Produce may not be “necessary” for a healthy diet. While most healthful diets include produce--and many of the world’s longest lived people eat predominantly plants--there are traditional cultures living almost exclusively on animal products (e.g., Inuit, Maasai) who enjoy excellent health.

- Line 10: the authors recognize that “access” has at least five distinct dimensions and then conclude (using the word “So”) that they should only focus on three of these dimensions? (ignoring the other two?) Also, it is not clear (in all instances beyond this point in the text) if “access” means “availability”, “accessibility”, “affordability”, or some combination(s). For example, Line 17: “car availability would also change access conditions.” So in this case, “access” would mean “accessibility” only, right?

- Line 21: the authors state they “attempted to control for it” but it seems more like they were using it as an effect modifier.

- Lines 22, 25, and elsewhere: what is “granted access”? On lines 31-32, the authors seem to imply it has something to do with being “owners” but it is not clear what they mean.

- Lines 29-30: it is hard to appreciate how “agriculture workers,” “farmers,” and “traditional market produce sellers” are all categorized as “produce sellers.” For example, the first two groups of workers might be field hands who sell *nothing* at all. The authors go on to say (on line 32) that 77% of “produce sellers” own the produce that they sell. So what about the other 23%? Why would the authors assume any “produce sellers” have “more accessibility” or “more affordability”? Where is the data? In the U.S., the exact opposite is true; farm workers often have the *least* access to produce and are the *most* food insecure (e.g., https://civileats.com/2011/09/26/hunger-in-the-fields/). The details on lines 29-34 represent the set-up for the entire study, but these ideas seem to be completely unfounded.

- Line 45: “FV access”, meaning what exactly?

- This whole introduction is odd. It reads a bit like a long abstract or detailed outline of the whole paper. I suggest moving some details into the abstract. Other details belong in the other (more usual) sections of the paper.

Background

- Again: “access” is mentioned several times and it is not clear that it means the same thing in all cases. The authors need to be clear what they mean.

- The authors also need to be clear if “FV” (fruits and vegetables) includes only fresh or also frozen, canned, dried, and jarred (and defend their choice).

- Lines 76-79: I have no idea what either of these two sentences mean: “Therefore, the lack of agreement regarding the behavioral effect of food deserts may enhance the need to analyze food accessibility beyond a food environmental condition. Food access is likely to result from

the interaction of a household characteristics and its food environment.”

Materials and Methods:

- this entire analysis is based on self-reported data? There is no discussion of item testing, validity, or reliability. There is no mention of sampling weights or how analyses accounted for the complex survey data. There are no details about participation or response rates. There is no discussion about missing data or how it was addressed.

Results:

- Line 170-171: regarding “The higher prevalence of hypertension in Table 1 can be explained by the higher salt intake shown in Table 2,” absolutely not! Cross-sectional correlations never *explain* anything. Moreover, salt but one contributor to blood pressure.

- Line 174: is “compensated” by? (not the right wording for distributions that just happen to be what they are)

- Tables: why is the comparison group for Table 2 the “rest of the country” (correct) whereas for Table 1 it is the “country,” presumably also including the “produce sellers (incorrect)?

- neither education nor income are ever defined; also, “low,” “medium”, and “high” are never explained or justified.

- Lines 185-186: if “compared with the rest of the population, produce sellers consume a similar amount of FV,” what does that say about “granted access” (whatever that means)? To me it suggests it doesn’t exist and this whole study is based on a faulty premise.

- Lines 208: “education significantly leads to …”? No! This is a cross-sectional study. There are no valid causal (or directional) conclusions.

- Lines 210-211: Again, not “leads to.” Also, presumably the effect of better “access” (whatever that means) would be mediated through greater consumption. But greater consumption is not seen so the lack of change of BMI is not really surprising unless the authors are postulating different potential mechanisms/pathways.

- Line 214 and 215: “treated group”? “control group”? This is a cross-sectional observational study. It is not a trial.

Discussion

- Lines 234-235: regarding, “a healthy food basket tends to be more expensive than an unhealthy one,” this statement can be challenged. Several studies show the monetary costs for healtfhful foods being cheaper (e.g., dried bean, lentils, grains). Also, it depend if you are looking per calorie, per weight, or by satiating potential. Additionally, some research has factored in the “time cost” of foods (time needed for soaking, chopping, cooking, etc.). The authors should specify what they mean by “expensive” in this regard.

- Line 238: it is not clear how the “underlying assumption” of subsidies, vouchers, and taxes would have anything to do with access “geographically.”

- Line 242: Height (in the absence of BMI) is not an “obesity-related indicator;” it may be an indicator of nutritional adequacy, but that is something different.

- Line 244: whatever this study is, it absolutely is not “a natural experiment of a free food environment.” Again, this is a cross-sectional, observations study, with cohorts (heterogeneous mix of “produce sellers” vs. everyone else) having food environments that might not be meaningfully different.

- Line 249: Another “leads to” (absolutely not!)

- Line 266: how can results be extrapolated to “food environments” (never defined) “where access to FV is granted or cheap”? Working on a farm (say picking potatoes) does not guarantee the field worker “access” (in any sense) to fruits or vegetables in general (or even to potatoes specifically). Working on a farm certainly does not mean workers are offered any part of harvests (cheaply, or at all). Without data, this presumption is just fundamentally flawed.

- Line 276: the authors found an association with level of schooling (presumably, although “education” never defined or explained). They did not find some benefit for any educational strategy and, moreover, have no data to support “likely to be effective at increasing FV consumption.” For one reason, produce consumption was not even an outcome in this study—and if it was, there was no difference between the groups! For another, as already mentioned repeatedly, this was a cross-sectional study! (and one based on self-reported data, the reliability and validity of which are never mentioned).

Reviewer #2: This paper presents data from a natural experiment that allows access to produce to be eliminated as a variable to explain differences in BMI among populations. This is a very creative way to test several hypothesis regarding the role of various factors in contributing to BMI. The data were thoroughly analyzed and the results were presented with clarity.

My only critique of the paper is with the writing style. The first three paragraphs of the introduction are choppy and, in my opinion, don't clearly provide context for the study. The first sentence states "Among healthy foods, produce holds a place of privilege." There are two unsupported propositions here - that some foods are always healthy and others are not, and consuming the healthy foods is a privilege. I think it would be best to avoid such judgements and say something like this: Fresh fruits and vegetables (FV) are a natural source of vitamins, minerals and fiber and are a critical component of a healthy diet. Literature suggests that a diet high in FVs leads to better health outcomes. However, consuming recommended amounts of FVs can be constrained by availability, accessibility, affordability, acceptability and accommodation. Previous studies have attempted to measure the role of food access on health indicators such as consumption of FVs per capita and BMI, but measuring access is problematic. People do not necessarily move in straight lines . . .

I wasn't sure what was meant by, "Supermarkets make healthy and unhealthy food more available" (76)

There are also unusual phrases, such as "literature has enhanced the role of access." (7, 14)

We assume this sample has granted access (30, 60)

which forces to redefine food maps (73)

by pursuing to provide (89)

weighted (217)

contributing to improve their diet (239)

After the introduction, the writing becomes better.

These comments are easily addressed.

Overall, the paper makes an important contribution to our understanding of obesity rates and where interventions would be most effective.

6. PLOS authors have the option to publish the peer review history of their article (what does this mean?). If published, this will include your full peer review and any attached files.

Reviewer #1: Yes: Sean C Lucan

Reviewer #2: No

---

## [Author Response · Author response to Decision Letter 0]

21 Jul 2020

Answers to Reviewer #1

Overall:

- Constructions like “According to [4]” (line 10) are quite odd. Other examples of similar constructions appear on all of the following lines: 65, 71, 74, 140, and 274.

Authors: Thank you. The complete manuscript was reviewed by an English teacher, Master of Arts in linguistics/proofreader taking special attention to your comments. 

Abstract:

- “access” meaning what? (even parenthetically) One or more of five dimensions discussed later?

- “education” defined how? (even parenthetically) Years of schooling? Subject-matter expertise?

- “income” defined how? (even parenthetically) For individuals / households? Absolute / relative?

- What are “produce sellers”? Farm-stand workers? Farmers? Produce-store owners? Fruit-cart vendors?

- What does “granted access” mean?

Authors: Thank you. We have revised the abstract to make explicit the terms that you mention. As requested, we have highlighted the added text and explicitly show the deleted text.

Intro:

- Line 2: Fruits and vegetables do not have to be “fresh” to be “produce.” Frozen, canned, dried, and jarred fruits and vegetables also count.

- Line 3: Produce may not be “necessary” for a healthy diet. While most healthful diets include produce--and many of the world’s longest lived people eat predominantly plants--there are traditional cultures living almost exclusively on animal products (e.g., Inuit, Maasai) who enjoy excellent health.

- Line 10: the authors recognize that “access” has at least five distinct dimensions and then conclude (using the word “So”) that they should only focus on three of these dimensions? (ignoring the other two?) Also, it is not clear (in all instances beyond this point in the text) if “access” means “availability”, “accessibility”, “affordability”, or some combination(s). For example, Line 17: “car availability would also change access conditions.” So in this case, “access” would mean “accessibility” only, right?

- Line 21: the authors state they “attempted to control for it” but it seems more like they were using it as an effect modifier.

- Lines 22, 25, and elsewhere: what is “granted access”? On lines 31-32, the authors seem to imply it has something to do with being “owners” but it is not clear what they mean.

- Lines 29-30: it is hard to appreciate how “agriculture workers,” “farmers,” and “traditional market produce sellers” are all categorized as “produce sellers.” For example, the first two groups of workers might be field hands who sell *nothing* at all. The authors go on to say (on line 32) that 77% of “produce sellers” own the produce that they sell. So what about the other 23%? Why would the authors assume any “produce sellers” have “more accessibility” or “more affordability”? Where is the data? In the U.S., the exact opposite is true; farm workers often have the *least* access to produce and are the *most* food insecure (e.g., https://civileats.com/2011/09/26/hunger-in-the-fields/). The details on lines 29-34 represent the set-up for the entire study, but these ideas seem to be completely unfounded.

Authors: Thank you. We have redefined the groups. Now, we do not take agricultural/farm workers into account. Close to 22% fruit and vegetable (FV) sellers do not own the products that they sell. In other words, they are employed by someone else to sell fruits and vegetables. 

- Line 45: “FV access”, meaning what exactly?

- This whole introduction is odd. It reads a bit like a long abstract or detailed outline of the whole paper. I suggest moving some details into the abstract. Other details belong in the other (more usual) sections of the paper.

Authors: Thank you. We have revised the introduction to make explicit the terms that you mention. We have worked to make it flow better than before. Following the journal guidelines for a review, we highlighted the changes in the manuscript. 

Background

- Again: “access” is mentioned several times and it is not clear that it means the same thing in all cases. The authors need to be clear what they mean.

- The authors also need to be clear if “FV” (fruits and vegetables) includes only fresh or also frozen, canned, dried, and jarred (and defend their choice).

- Lines 76-79: I have no idea what either of these two sentences mean: “Therefore, the lack of agreement regarding the behavioral effect of food deserts may enhance the need to analyze food accessibility beyond a food environmental condition. Food access is likely to result from the interaction of a household characteristics and its food environment.”

Authors: Thank you. We have revised this section to highlight access definition and reworded the mentioned statement. Following the journal guidelines for a review, we highlighted the changes in the manuscript. 

Materials and Methods:

- this entire analysis is based on self-reported data? There is no discussion of item testing, validity, or reliability. There is no mention of sampling weights or how analyses accounted for the complex survey data. There are no details about participation or response rates. There is no discussion about missing data or how it was addressed.

Authors: We have included in the data section a more detailed description about the way the dataset is built, including information on the sampling weights and response rate.

Results:

- Line 170-171: regarding “The higher prevalence of hypertension in Table 1 can be explained by the higher salt intake shown in Table 2,” absolutely not! Cross-sectional correlations never *explain* anything. Moreover, salt but one contributor to blood pressure.

Authors: Thank you. We have changed that part. 

- Line 174: is “compensated” by? (not the right wording for distributions that just happen to be what they are)

Authors: We agree and eliminated this wording.

- Tables: why is the comparison group for Table 2 the “rest of the country” (correct) whereas for Table 1 it is the “country,” presumably also including the “produce sellers (incorrect)?

Authors: Thank you for this comment. We remade Table 1 with rest of the country for consistency between tables 1 and 2.

- neither education nor income are ever defined; also, “low,” “medium”, and “high” are never explained or justified.

Authors: Thanks for pointing at that. We defined each category within Table 2.

- Lines 185-186: if “compared with the rest of the population, produce sellers consume a similar amount of FV,” what does that say about “granted access” (whatever that means)? To me it suggests it doesn’t exist and this whole study is based on a faulty premise.

Authors: We assume that FFV sellers have more access than the rest of the population. This assumption is based on the fact that farmers and traditional market FFV sellers, in general, are also owners. In fact, 77% of traditional market FFV sellers own the produce they sell (SERCOTEC, 2016). We eliminated agricultural workers that were categorized as FFV sellers from our sample, so that currently only farmers and traditional market FFV sellers are categorized as FFV sellers.

More access does not necessarily mean more FV consumption, and this is something that the food desert literature has made an effort in showing but results vary greatly. Under the assumption that FV sellers have more access that the rest of the population, in our sample, they do not consume more FV and are more obese. This is precisely the puzzle that made us investigate our research question. Our answer to this puzzle is mainly the lack of education. We believe this is an important contribution of this paper.

- Lines 208: “education significantly leads to …”? No! This is a cross-sectional study. There are no valid causal (or directional) conclusions.

Authors: We would like to thank the reviewer for this comment. We soften our language throughout the paper regarding causality. We understand that in medical research cross-sectional studies may be used to describe some feature of the population but cannot be used to show a causal effect. In the field of economics, “the notion of ceteris paribus—that is, holding all other (relevant) factors fixed—is at the crux of establishing a causal relationship. Simply finding that two variables are correlated is rarely enough to conclude that a change in one variable causes a change in another. After all, rarely can we run a controlled experiment that allows a simple correlation analysis to uncover causality. Instead, we can use econometric methods to effectively hold other factors fixed.” (Wooldridge, 2010, p.3). Therefore, we are using a matching technique in an attempt to provide a counterfactual. The ideal methodology would be to conduct a randomized experiment. So, this is a limitation of our paper. We added a paragraph in the discussion mentioning the limitations of our analysis, especially the cross-sectional nature of our data.

- Lines 210-211: Again, not “leads to.” Also, presumably the effect of better “access” (whatever that means) would be mediated through greater consumption. But greater consumption is not seen so the lack of change of BMI is not really surprising unless the authors are postulating different potential mechanisms/pathways.

Authors: Instead of “leads to” we have changed the wording to “associated with.”

The point of FV seller better access and FV consumption is responded above.

- Line 214 and 215: “treated group”? “control group”? This is a cross-sectional observational study. It is not a trial.

Authors: We would like to thank the reviewer for this comment. In fact, we do not conduct an experiment but we do conduct a matching procedure making an attempt to find a counterfactual group for FFV sellers. The names “treatment” and “control” stem from the first applications of these techniques but nowadays applications vary widely (Wooldridge, 2010). A part of the matching procedure is assigning a group as the “treatment” and a group as the “control.” So, based on the reviewer’s comment we added a footnote clarifying that we do not conduct an experiment.

Discussion

- Lines 234-235: regarding, “a healthy food basket tends to be more expensive than an unhealthy one,” this statement can be challenged. Several studies show the monetary costs for healtfhful foods being cheaper (e.g., dried bean, lentils, grains). Also, it depend if you are looking per calorie, per weight, or by satiating potential. Additionally, some research has factored in the “time cost” of foods (time needed for soaking, chopping, cooking, etc.). The authors should specify what they mean by “expensive” in this regard.

Authors: Thanks for pointing at that. We clarified that we refer to “expensive” in terms of energy density. We added a reference and an explanation regarding the relationship between energy density and energy cost.

- Line 238: it is not clear how the “underlying assumption” of subsidies, vouchers, and taxes would have anything to do with access “geographically.”

Authors: That paragraph was confusing, thank you for pointing it out. We rearranged and rephrased it to be more consistent with changes made in the introduction regarding accessibility and affordability.

- Line 242: Height (in the absence of BMI) is not an “obesity-related indicator;” it may be an indicator of nutritional adequacy, but that is something different.

Authors: Yes, thank you for this comment. We decided to maintain height as part of our analyses because it helps in the BMI interpretation, for example, if weight increases and BMI does not, it is easier to understand what is happening if we observe that height also increases. If the reviewer thinks we should eliminate height from the analyses, please let us know and we can eliminate it. Also, we added a footnote in the materials and methods section to clarify that we don’t use height as an obesity indicator.

- Line 244: whatever this study is, it absolutely is not “a natural experiment of a free food environment.” Again, this is a cross-sectional, observations study, with cohorts (heterogeneous mix of “produce sellers” vs. everyone else) having food environments that might not be meaningfully different.

Authors: Point taken. We eliminated the phrase “a natural experiment of a free food environment.”

- Line 249: Another “leads to” (absolutely not!)

Authors: Point taken. We changed “leads to” to “associated with.”

- Line 266: how can results be extrapolated to “food environments” (never defined) “where access to FV is granted or cheap”? Working on a farm (say picking potatoes) does not guarantee the field worker “access” (in any sense) to fruits or vegetables in general (or even to potatoes specifically). Working on a farm certainly does not mean workers are offered any part of harvests (cheaply, or at all). Without data, this presumption is just fundamentally flawed.

Authors: Thank you for this comment. We agree and eliminated agricultural workers as part of the FFV sellers’ group and re-estimated all our tables.

- Line 276: the authors found an association with level of schooling (presumably, although “education” never defined or explained). They did not find some benefit for any educational strategy and, moreover, have no data to support “likely to be effective at increasing FV consumption.” For one reason, produce consumption was not even an outcome in this study—and if it was, there was no difference between the groups! For another, as already mentioned repeatedly, this was a cross-sectional study! (and one based on self-reported data, the reliability and validity of which are never mentioned).

Authors: Thank you for pointing at that. We now define education and use years of education (schooling) throughout the paper. Also, we replaced “FV consumption” with “obesity indicators” in the discussion section.

Answers to Reviewer #2

This paper presents data from a natural experiment that allows access to produce to be eliminated as a variable to explain differences in BMI among populations. This is a very creative way to test several hypotheses regarding the role of various factors in contributing to BMI. The data were thoroughly analyzed and the results were presented with clarity.

My only critique of the paper is with the writing style. The first three paragraphs of the introduction are choppy and, in my opinion, don't clearly provide context for the study. The first sentence states "Among healthy foods, produce holds a place of privilege." There are two unsupported propositions here - that some foods are always healthy and others are not, and consuming the healthy foods is a privilege. I think it would be best to avoid such judgements and say something like this: Fresh fruits and vegetables (FV) are a natural source of vitamins, minerals and fiber and are a critical component of a healthy diet. Literature suggests that a diet high in FVs leads to better health outcomes. However, consuming recommended amounts of FVs can be constrained by availability, accessibility, affordability, acceptability and accommodation. Previous studies have attempted to measure the role of food access on health indicators such as consumption of FVs per capita and BMI, but measuring access is problematic. People do not necessarily move in straight lines . . .

Authors: Thank you. We use the suggested paragraph to start improving the introduction. Following the journal guidelines for a review, we highlighted the changes in the manuscript.

I wasn't sure what was meant by, "Supermarkets make healthy and unhealthy food more available" (76)

Authors: Thank you. We have edited it.

There are also unusual phrases, such as "literature has enhanced the role of access." (7, 14)

Authors: Thank you. We have edited it.

We assume this sample has granted access (30, 60)

Authors: Thank you. We have edited it. We assume this sample has granted access (30, 60) a guaranteed access.

which forces to redefine food maps (73)

Authors: Thank you. We have changed to “which would imply redefining food maps.” 

by pursuing to provide (89)

Authors: Thank you. We have changed to “by providing”.

weighted (217)

Authors: Thank you. We have deleted because it’s unnecessary.

contributing to improve their diet (239)

Authors: Thank you. We have deleted because it’s implicit. 

After the introduction, the writing becomes better.

Authors: Thank you. We have edited the all mentioned statements. Following the journal guidelines for a review, we highlighted the changes in the manuscript.

These comments are easily addressed.

Overall, the paper makes an important contribution to our understanding of obesity rates and where interventions would be most effective.

---

## [Decision Letter · Decision Letter 1]

18 Sep 2020

PONE-D-20-02163R1

Obesity under Full Fresh Fruit and Vegetable Access Conditions

PLOS ONE

Dear Dr. Silva,

Thank you for submitting your manuscript to PLOS ONE. After careful consideration, we feel that it has merit but does not fully meet PLOS ONE’s publication criteria as it currently stands. Therefore, we invite you to submit a revised version of the manuscript that addresses the points raised during the review process.

The revised version should take into account all the comments in the reports.

We look forward to receiving your revised manuscript.

Kind regards,

Petri Böckerman

Academic Editor

PLOS ONE

Reviewers' comments:

Reviewer's Responses to Questions

**Comments to the Author**

1. If the authors have adequately addressed your comments raised in a previous round of review and you feel that this manuscript is now acceptable for publication, you may indicate that here to bypass the “Comments to the Author” section, enter your conflict of interest statement in the “Confidential to Editor” section, and submit your "Accept" recommendation.

Reviewer #2: All comments have been addressed

Reviewer #3: (No Response)

2. Is the manuscript technically sound, and do the data support the conclusions?

Reviewer #2: Yes

Reviewer #3: No

3. Has the statistical analysis been performed appropriately and rigorously? 

Reviewer #2: Yes

Reviewer #3: No

4. Have the authors made all data underlying the findings in their manuscript fully available?

Reviewer #2: Yes

Reviewer #3: Yes

5. Is the manuscript presented in an intelligible fashion and written in standard English?

Reviewer #2: Yes

Reviewer #3: Yes

6. Review Comments to the Author

Reviewer #2: The authors did a good job addressing the previous concerns, and the writing style was greatly improved.

Reviewer #3: Referee report on "Obesity under full fresh fruit and vegetable access conditions" (PONE-D-20-02163R1)

In this paper, the authors examined how better access to fresh fruit and vegetables (FFV) is associated with BMI using cross-section data. The identification strategy is based on the assumption that farmers and traditional market sellers presumably have better access to FFV. Estimations employ OLS, matching, and instrumental variables methods. I think the authors have addressed most of the comments reviewers have pointed out. However, I still have some major concerns related to estimation methods and interpretations.

1. The authors now use an indicator variable for FFV sellers to explain BMI. However, the idea of the paper is that FFV sellers have better access to FFC products, which may affect their FFV consumption, and therefore, BMI. So, instead of estimating a reduced form model, as they do now, they should estimate a mediation model which would answer to three questions: 1) Is better access to FFV linked with FFV consumption (based on Table 2, you have information on FFV consumption), 2) Is FFV consumption associated with BMI, and 3) does FFV consumption mediate the potential link between access to FFV and BMI. Using a mediation model with adequate control variables (e.g. education, income, and sex), you could do that (see e.g. sgmediation command in STATA).

If you find that being an FFV seller is linked with higher consumption of FFV products (with adequate controls; results in Table 2 do not control e.g. for education and income), that may also support your argument that FFV sellers have better access to FFV products (R1 seemed to be concerned whether FFV sellers actually have better access).

Now the authors find, based on OLS and matching results (I am quite skeptical about your IV results, I will explain below), that being an FFV seller is associated with higher BMI, which is a bit surprising and needs an explanation. I think that with a mediation model they maybe could find an explanation to this finding.

2. Height as an outcome variable. I think this is not a relevant outcome variable in this paper. Instead of using it as an outcome variable, I would use it as a control variable in the models that use weight as the outcome variable.

3. I am skeptical of the IV results. The authors use the mother’s years of educations and its square as instruments for a child’s education. Intuitively, the same unobservable factors may explain both mother’s and child’s education years, which violates the IV assumptions. Also, Sargan’s test of overidentifying restrictions (Table 8) rejects the null hypothesis (p-value in the BMI equation, i.e. in your main results, was 0.043), which implies that the instruments (as a group) are not exogenous. I would not report these results in the paper.

4. Please be explicit with your data description in section 2.2: What year(s) does your data (i.e. the data that you use in the estimations) cover? Which variables are self-reported and which are based on measurements conducted by health care professionals? For example, is BMI self-reported or not?

5. There is a detailed description of the matching model but no discussion about IV method and instrument validity. However, as said, I am skeptical that your instrument is valid so, I would drop those results.

6. In Table 2 you use t-test to indicate whether the differences are statistically significant. Why don’t you also show similar results in Table 1?

7. On page 7, the authors state: “the variable FFV seller shows significant effects on weight and height that may explain the weak effect on BMI”. I don’t understand this statement.

8. The authors conclude that “years of education, more than household income, is associated with BMI reductions”. I don’t see where this interpretation comes from.

9. In the Discussion section: “We used the ENS in Chile to assess the change on FFV sellers’ obesity related indicators”. The way I understood the setup, the authors explain BMI levels, not changes.

10. Table 3: In column 3, the authors use population weights. If sample weights are necessary, why don’t you use them in the OLS and matching models? Also, if weights are necessary, please explain that in the section “Materials and Methods”.

11. On page 3 (last paragraph), there are two times “second” (should be second and third). Therefore it is unclear to which point the authors refer when they say that “we cannot test the second condition directly”.

7. PLOS authors have the option to publish the peer review history of their article (what does this mean?). If published, this will include your full peer review and any attached files.

Reviewer #2: No

Reviewer #3: No

---

## [Author Response · Author response to Decision Letter 1]

12 Nov 2020

1. The authors now use an indicator variable for FFV sellers to explain BMI. However, the idea of the paper is that FFV sellers have better access to FFC products, which may affect their FFV consumption, and therefore, BMI. So, instead of estimating a reduced form model, as they do now, they should estimate a mediation model which would answer to three questions: 1) Is better access to FFV linked with FFV consumption (based on Table 2, you have information on FFV consumption), 2) Is FFV consumption associated with BMI, and 3) does FFV consumption mediate the potential link between access to FFV and BMI. Using a mediation model with adequate control variables (e.g. education, income, and sex), you could do that (see e.g. sgmediation command in STATA).

If you find that being an FFV seller is linked with higher consumption of FFV products (with adequate controls; results in Table 2 do not control e.g. for education and income), that may also support your argument that FFV sellers have better access to FFV products (R1 seemed to be concerned whether FFV sellers actually have better access).

Now the authors find, based on OLS and matching results (I am quite skeptical about your IV results, I will explain below), that being an FFV seller is associated with higher BMI, which is a bit surprising and needs an explanation. I think that with a mediation model they maybe could find an explanation to this finding.

Answer: Thank you. We have replaced the OLS results (including IV OLS) by the mediation results. The mediation results are also consistent with the matching results. FFV sellers, despite having easy access to FFV, have a higher weight and BMI. As expected, higher education and higher income are associated with lower weight and BMI.

2. Height as an outcome variable. I think this is not a relevant outcome variable in this paper. Instead of using it as an outcome variable, I would use it as a control variable in the models that use weight as the outcome variable.

Answer: Thank you. We have done it that way. We agree that it makes sense to include height as an independent variable in weight models. 

3. I am skeptical of the IV results. The authors use the mother’s years of educations and its square as instruments for a child’s education. Intuitively, the same unobservable factors may explain both mother’s and child’s education years, which violates the IV assumptions. Also, Sargan’s test of overidentifying restrictions (Table 8) rejects the null hypothesis (p-value in the BMI equation, i.e. in your main results, was 0.043), which implies that the instruments (as a group) are not exogenous. I would not report these results in the paper.

Answer: Thank you. We have replaced the IV OLS estimation by the mediation model results.

4. Please be explicit with your data description in section 2.2: What year(s) does your data (i.e. the data that you use in the estimations) cover? Which variables are self-reported and which are based on measurements conducted by health care professionals? For example, is BMI self-reported or not?

Answer: Thank you for this comment. We modified the “Data” subsection including the points requested. Specifically, we included: survey year used in our analyses, self-reported variables and those variables measured by health care professionals.

5. There is a detailed description of the matching model but no discussion about IV method and instrument validity. However, as said, I am skeptical that your instrument is valid so, I would drop those results.

Answer: Thank you. We have replaced the IV OLS estimation by the mediation model results.

6. In Table 2 you use t-test to indicate whether the differences are statistically significant. Why don’t you also show similar results in Table 1?

Answer: Thank you. We have included this analysis.

7. On page 7, the authors state: “the variable FFV seller shows significant effects on weight and height that may explain the weak effect on BMI”. I don’t understand this statement.

Answer: Thanks for this comment. We have deleted the regression on height as suggested, so the phrase mentioned in this comment was also deleted.

8. The authors conclude that “years of education, more than household income, is associated with BMI reductions”. I don’t see where this interpretation comes from.

Answer: Thank you. We have changed this phrase to incorporate the importance of both high education levels and high income levels as BMI reducers. 

9. In the Discussion section: “We used the ENS in Chile to assess the change on FFV sellers’ obesity related indicators”. The way I understood the setup, the authors explain BMI levels, not changes.

Answer: That was a mistake, thanks for pointing at it.

10. Table 3: In column 3, the authors use population weights. If sample weights are necessary, why don’t you use them in the OLS and matching models? Also, if weights are necessary, please explain that in the section “Materials and Methods”.

Answer: Thank you. We have dropped the population weights from the estimation. We used them only for the descriptive statistics.

11. On page 3 (last paragraph), there are two times “second” (should be second and third). Therefore it is unclear to which point the authors refer when they say that “we cannot test the second condition directly”.

Answer: Yes, that was a mistake. Thank you for catching it. Now it says third instead of second. We have also eliminated that a condition could not be tested, as all of them could eventually be tested for. However, it seems there isn’t a validated test for unconfoundedness available yet. We do show supporting evidence for conditions one (overlap) and two (balanced sample) in the Appendix.

---

## [Decision Letter · Decision Letter 2]

27 Nov 2020

PONE-D-20-02163R2

Obesity under Full Fresh Fruit and Vegetable Access Conditions

PLOS ONE

Dear Dr. Silva,

Thank you for submitting your manuscript to PLOS ONE. After careful consideration, we feel that it has merit but does not fully meet PLOS ONE’s publication criteria as it currently stands. Therefore, we invite you to submit a revised version of the manuscript that addresses the points raised during the review process.

Reviewer #3 still has very serious concerns regarding your revised version of the paper. I encourage that you revise the paper only if you can address all her/his concerns.

We look forward to receiving your revised manuscript.

Kind regards,

Petri Böckerman

Academic Editor

PLOS ONE

Reviewers' comments:

Reviewer's Responses to Questions

**Comments to the Author**

1. If the authors have adequately addressed your comments raised in a previous round of review and you feel that this manuscript is now acceptable for publication, you may indicate that here to bypass the “Comments to the Author” section, enter your conflict of interest statement in the “Confidential to Editor” section, and submit your "Accept" recommendation.

Reviewer #2: All comments have been addressed

Reviewer #3: (No Response)

2. Is the manuscript technically sound, and do the data support the conclusions?

Reviewer #2: Yes

Reviewer #3: Partly

3. Has the statistical analysis been performed appropriately and rigorously? 

Reviewer #2: Yes

Reviewer #3: No

4. Have the authors made all data underlying the findings in their manuscript fully available?

Reviewer #2: Yes

Reviewer #3: Yes

5. Is the manuscript presented in an intelligible fashion and written in standard English?

Reviewer #2: Yes

Reviewer #3: Yes

6. Review Comments to the Author

Reviewer #2: There were just a few editorial issues.

1) Third sentence of the abstract should read, "Using fruit and vegetable access as a mediator, we found that years of education and household income are correlated with a decrease in obesity." (Add "access" and remove comma.)

2) Line 13; "agricultural farmers" is redundant. Just say "farmers." I think this occurred in one other place as well.

3) Sentence on lines 70 - 72 is unclear. ". . . socio-demographic characteristics lead to a food consumption pattern or the opposite." Unclear what "opposite" means in this context.

4) Lines 115 and 275 - comma is unnecessary

5) Table 2 - acceptable abbreviation for grams is "g" not "gr"

6) Table 3 - "height" is misspelled

7) Line 294 - "to" should be "with"

The authors are to be congratulated on a fine paper.

Reviewer #3: The authors have responded adequately to many of my comments but still, have major concerns regarding the paper.

1. In the introduction, the authors refer to the results from the mediator model solely saying “using mediation regression, we found that years of education and household income are associated with a smaller BMI”. This is a bit odd because this is not the main point why you performed a mediation analysis.

2. Mediation model: The authors refer to Baron & Kenny, who suggest that mediation analysis should be done in three steps. The paper by Iacobucci et al. (2007, p.153), to which the authors also refer, proposes conducting the mediation analysis via SEM. Whether mediation is significant is typically also tested using e.g., the Sobel test. It seems that this is not what the authors have done. In Table 3, they estimate two regressions separately.

3. Interpretation of the mediation results: The idea of the mediation model is to see 1) whether the independent variable (being an FFV seller) is associated with the mediator (FV consumption); 2) whether the mediator (FV consumption) is associated with the outcome (BMI) and; 3) whether this mediated pathway (1 and 2) is significant. The “FFV seller” coefficients in Table 3 (Columns 2 and 4) are related to point 1, and the “FV portions” coefficients in Table 3 (Columns 3 and 5) are related to point 2. The authors do not comment at all the aforementioned “FFV seller” coefficients in columns 2 and 4. Then, based solely on the “FV portions” coefficient, the authors conclude that “FV consumption is not acting as a mediator to explain BMI variation”. In some sense that applies, because for the mediation pathway to be significant, there needs to be a significant relationship between the independent variable and mediator and between mediator and outcome variable. However, making this interpretation solely based on the association between mediator and outcome variable is odd. E.g., the Sobel test could be used to test the significance.

4. Findings: The authors find that FFC sellers have higher BMI/weight. Why? It is hard for me to understand why simply being an FFV seller is associated with higher BMI unless there are unobserved confounders that affect the results. One potential explanation for this finding would have been that because FFV sellers have better access to FV products, they consume more calories (i.e., they consume FV products on top of all other food products). However, the mediation model results imply that this is not the case: FFV sellers do not seem to consume more FV products, and FV product consumption does not seem to be associated with BMI.

5. Conclusions: The main finding from the mediation model should be stated more clearly. Based on the results, the main finding seems to be that having better FFV access does not reduce BMI because better access is not associated with FFV consumption and because FFV consumption is not associated with BMI. There is no discussion about the finding that FFV consumption is not associated with weight. Does this finding receive support from other studies? From the point of view that FFV consumption is not associated with weight, some parts in the “Discussion” section seem a bit odd.

6. Causal terminology: The authors use causal terminology (e.g., “the fifth column shows the effect on weight…”), although the methods they use are usually not considered to allow causal interpretation.

7. Limitations of the study: The limitations of the study and the implications of these limitations require more discussion. For example, 1) the number of FFV sellers is rather small (96). This is a bit unfortunate because there is some indication that FFV sellers consume more FV products, but the differences are not significant. 2) Although the authors argue otherwise, it is possible that FFV sellers actually do not have better access to FV products. The argument that FFV sellers have better access to FV products, cannot be tested directly. If the authors would have found that FFV sellers consumed more FV products, that might have supported their argument. 3) Potential confounders (see my previous point 4).

8. At some point, I got lost with abbreviations FFV and FV. The authors talk e.g., about FFV access and FFV baskets but also about FV consumption. Would one abbreviation be enough?

7. PLOS authors have the option to publish the peer review history of their article (what does this mean?). If published, this will include your full peer review and any attached files.

Reviewer #2: No

Reviewer #3: No

---

## [Author Response · Author response to Decision Letter 2]

13 Jan 2021

Answers to Reviewer #2

1) Third sentence of the abstract should read, "Using fruit and vegetable access as a mediator, we found that years of education and household income are correlated with a decrease in obesity." (Add "access" and remove comma.)

2) Line 13; "agricultural farmers" is redundant. Just say "farmers." I think this occurred in one other place as well.

3) Sentence on lines 70 - 72 is unclear. ". . . socio-demographic characteristics lead to a food consumption pattern or the opposite." Unclear what "opposite" means in this context.

4) Lines 115 and 275 - comma is unnecessary

5) Table 2 - acceptable abbreviation for grams is "g" not "gr"

6) Table 3 - "height" is misspelled

7) Line 294 - "to" should be "with"

Authors: Thank you. We have changed all the above comments made by the reviewer. 

The authors are to be congratulated on a fine paper.

Authors: Thank you. We appreciate the constructive feedback over the review process. 

Answers to Reviewer #3

1. In the introduction, the authors refer to the results from the mediator model solely saying “using mediation regression, we found that years of education and household income are associated with a smaller BMI”. This is a bit odd because this is not the main point why you performed a mediation analysis.

Authors: Thanks for pointing at that. We have edited the paragraph to make it more informative (the changes are highlighted in the text).

2. Mediation model: The authors refer to Baron & Kenny, who suggest that mediation analysis should be done in three steps. The paper by Iacobucci et al. (2007, p.153), to which the authors also refer, proposes conducting the mediation analysis via SEM. Whether mediation is significant is typically also tested using e.g., the Sobel test. It seems that this is not what the authors have done. In Table 3, they estimate two regressions separately.

Authors: In the lines 255 to 259, we have added an additional explanation regarding the estimation procedure and the Sobel test results are available in the appendix.

3. Interpretation of the mediation results: The idea of the mediation model is to see 1) whether the independent variable (being an FFV seller) is associated with the mediator (FV consumption); 2) whether the mediator (FV consumption) is associated with the outcome (BMI) and; 3) whether this mediated pathway (1 and 2) is significant. The “FFV seller” coefficients in Table 3 (Columns 2 and 4) are related to point 1, and the “FV portions” coefficients in Table 3 (Columns 3 and 5) are related to point 2. The authors do not comment at all the aforementioned “FFV seller” coefficients in columns 2 and 4. Then, based solely on the “FV portions” coefficient, the authors conclude that “FV consumption is not acting as a mediator to explain BMI variation”. In some sense that applies, because for the mediation pathway to be significant, there needs to be a significant relationship between the independent variable and mediator and between mediator and outcome variable. However, making this interpretation solely based on the association between mediator and outcome variable is odd. E.g., the Sobel test could be used to test the significance.

Authors: We recognize that the previous discussion needed to be refined. In the line 255 to 259, we have added further discussion and support based on the Sobel test. As presented in the Appendix, the Sobel test results are aligned with our results of no mediation path through FV consumption.

4. Findings: The authors find that FFC sellers have higher BMI/weight. Why? It is hard for me to understand why simply being an FFV seller is associated with higher BMI unless there are unobserved confounders that affect the results. One potential explanation for this finding would have been that because FFV sellers have better access to FV products, they consume more calories (i.e., they consume FV products on top of all other food products). However, the mediation model results imply that this is not the case: FFV sellers do not seem to consume more FV products, and FV product consumption does not seem to be associated with BMI.

Authors: Thank you for pointing at that. Based on our analysis, fresh FV sellers have a weight and BMI similar to people with the same education level. Columns 4 and 5 in Table 3 show that weight and BMI are not significantly different comparing fresh FV sellers and the rest of the population when using a low-education subsample. Therefore, the variations on weight and BMI are associated with changes on education rather than with being a fresh FV seller. We understand that we need to make this more explicit in the discussion section.

5. Conclusions: The main finding from the mediation model should be stated more clearly. Based on the results, the main finding seems to be that having better FFV access does not reduce BMI because better access is not associated with FFV consumption and because FFV consumption is not associated with BMI. There is no discussion about the finding that FFV consumption is not associated with weight. Does this finding receive support from other studies? From the point of view that FFV consumption is not associated with weight, some parts in the “Discussion” section seem a bit odd.

Authors: Thank you. We have incorporated your comments in the discussion section and stated the mediation results more explicitly. We also included a discussion on our finding relating FV consumption and weight and added references that support our findings. 

6. Causal terminology: The authors use causal terminology (e.g., “the fifth column shows the effect on weight…”), although the methods they use are usually not considered to allow causal interpretation.

Authors: Thank you for this comment. We did our best to eliminate any causal interpretations/terminology of our results. 

7. Limitations of the study: The limitations of the study and the implications of these limitations require more discussion. For example, 1) the number of FFV sellers is rather small (96). This is a bit unfortunate because there is some indication that FFV sellers consume more FV products, but the differences are not significant. 2) Although the authors argue otherwise, it is possible that FFV sellers actually do not have better access to FV products. The argument that FFV sellers have better access to FV products, cannot be tested directly. If the authors would have found that FFV sellers consumed more FV products, that might have supported their argument. 3) Potential confounders (see my previous point 4).

Authors: Yes, thank you for this comment. We have expanded the limitations’ paragraph considering your suggestions. We did not mention potential confounders explicitly in the limitations as we have already said that having cross sectional data is a limitation and we believe this implies potential omitted variables that could be correlated with independent and dependent variables. Along these lines, we also added that the survey was not originally designed to measure fresh FV sellers’ lifestyle.

8. At some point, I got lost with abbreviations FFV and FV. The authors talk e.g., about FFV access and FFV baskets but also about FV consumption. Would one abbreviation be enough?

Authors: Point taken. We have kept “FV”. When we need to refer to fresh fruit and vegetables, we indicate “fresh FV”.

Thank you for your comments.

---

## [Decision Letter · Decision Letter 3]

11 Feb 2021

PONE-D-20-02163R3

Obesity under Full Fresh Fruit and Vegetable Access Conditions

PLOS ONE

Dear Dr. Silva,

Thank you for submitting your manuscript to PLOS ONE. After careful consideration, we feel that it has merit but does not fully meet PLOS ONE’s publication criteria as it currently stands. Therefore, we invite you to submit a revised version of the manuscript that addresses the points raised during the review process.

The revised version should take into account all the remaining comments stated in the reports.

We look forward to receiving your revised manuscript.

Kind regards,

Petri Böckerman

Academic Editor

PLOS ONE

Reviewers' comments:

Reviewer's Responses to Questions

**Comments to the Author**

1. If the authors have adequately addressed your comments raised in a previous round of review and you feel that this manuscript is now acceptable for publication, you may indicate that here to bypass the “Comments to the Author” section, enter your conflict of interest statement in the “Confidential to Editor” section, and submit your "Accept" recommendation.

Reviewer #2: All comments have been addressed

Reviewer #3: (No Response)

2. Is the manuscript technically sound, and do the data support the conclusions?

Reviewer #2: (No Response)

Reviewer #3: Partly

3. Has the statistical analysis been performed appropriately and rigorously? 

Reviewer #2: (No Response)

Reviewer #3: I Don't Know

4. Have the authors made all data underlying the findings in their manuscript fully available?

Reviewer #2: (No Response)

Reviewer #3: Yes

5. Is the manuscript presented in an intelligible fashion and written in standard English?

Reviewer #2: (No Response)

Reviewer #3: Yes

6. Review Comments to the Author

Reviewer #2: (No Response)

Reviewer #3: Referee report on "Obesity under full fresh fruit and vegetable access conditions" (PONE-D-20-02163R1).

I still have major concerns regarding the paper, particularly regarding the mediation analysis.

1. The main results should be presented more clearly throughout the paper (abstract, intro, results, and conclusions).

2. The authors do not explain in the paper how the mediated path is obtained/calculated.

3. I do not understand Table 5. Based on Table 3, the indirect effect in the weight model is (0.397*0.0291) 0.021. However, based on Table 5 it is 0.000. You can do the Sobel test using the delta method. Why are there separate columns for Sobel and delta? Based on table notes the standard errors in all columns (delta, sobel, monte carlo) are based on bootstrapping?

4. Tables should be self-standing. In Table 4, it is unclear to which variable the coefficients refer to.

5. The first paragraph of section 2 (page 3): I do not think the description of the mediation model really captures the essence of the method.

6. The second paragraph in section 2.2: “Fresh FV sellers’ access can directly affect BMI…” I would put this just “being a fresh FV seller”. Access to me seems to refer to the mediation pathway.

7. The authors should be more careful with causal terminology. E.g., “We found that having better fresh FV access does not reduce BMI”. “… education leads to a stronger effect…”

8. Relative to the methods and findings of this paper, I find the conclusion that “education needs to be part with a more comprehensive public policy” a bit excessive.

9. I did not understand the following sentence in the Discussion section: “Since our results show that fresh FV sellers consume similar amounts of FV compared to the rest of the population, we do not have direct evidence… …. who consume more FV and have lower BMI).”

10. As a limitation, the authors mention that FV sellers could understate their true income. How is that a major limitation?

7. PLOS authors have the option to publish the peer review history of their article (what does this mean?). If published, this will include your full peer review and any attached files.

Reviewer #2: No

Reviewer #3: No

---

## [Author Response · Author response to Decision Letter 3]

11 Mar 2021

Answers to Reviewer #3

I still have major concerns regarding the paper, particularly regarding the mediation analysis.

1. The main results should be presented more clearly throughout the paper (abstract, intro, results, and conclusions).

Authors: Thanks for pointing at that. We have edited the four sections that you mentioned to highlight the results and explain them more clearly (the changes are highlighted in the text). 

2. The authors do not explain in the paper how the mediated path is obtained/calculated.

Authors: Thank you. We have incorporated an explanation in the Methodology section and we explain the computation in the Results section (the changes are highlighted in the text).

3. I do not understand Table 5. Based on Table 3, the indirect effect in the weight model is (0.397*0.0291) 0.021. However, based on Table 5 it is 0.000. You can do the Sobel test using the delta method. Why are there separate columns for Sobel and delta? Based on table notes the standard errors in all columns (delta, sobel, monte carlo) are based on bootstrapping?

Authors: We have corrected the Stata code. Now, Tables 3 and 5 present unstandardized coefficients, therefore, both tables are consistent (for weight equation: 0.3971*0.0291=0.012). To the best of our knowledge, Delta Method can be used or not to obtain the standard errors, which explain the two columns. Now, to avoid confusion, we have chosen to drop the Delta Method column, whose results are very similar to the other two test results. We have corrected the table notes, the bootstrapping is only for the Monte Carlo approach.

4. Tables should be self-standing. In Table 4, it is unclear to which variable the coefficients refer to.

Authors: Thanks for pointing at that. We have edited the footnote of Table 4 to make it more self-standing.

5. The first paragraph of section 2 (page 3): I do not think the description of the mediation model really captures the essence of the method.

Authors: Yes, thank you for this comment. We have edited the explanation including two references. The explanation of the mediation model is at the beginning of section 2 and continues in section 2.2.

6. The second paragraph in section 2.2: “Fresh FV sellers’ access can directly affect BMI…” I would put this just “being a fresh FV seller”. Access to me seems to refer to the mediation pathway.

Authors: Thanks for pointing at that. We have corrected it (the changes are highlighted in the text). 

7. The authors should be more careful with causal terminology. E.g., “We found that having better fresh FV access does not reduce BMI”. “… education leads to a stronger effect…”

Authors: Point taken. We have reviewed the causal methodology in the full article and hope to have eliminated all the causal terminology (the changes are highlighted in the text). 

8. Relative to the methods and findings of this paper, I find the conclusion that “education needs to be part with a more comprehensive public policy” a bit excessive.

Authors: We recognize that the previous statement was too strong. We have softened it. (The changes are highlighted in the text). 

9. I did not understand the following sentence in the Discussion section: “Since our results show that fresh FV sellers consume similar amounts of FV compared to the rest of the population, we do not have direct evidence… …. who consume more FV and have lower BMI).”

Authors: Thanks for pointing at that. We have eliminated the statement (the changes are highlighted in the text).

10. As a limitation, the authors mention that FV sellers could understate their true income. How is that a major limitation?

Authors: We believed that as income is one of our control variables, and it is not directly observed, we cannot completely rely on how accurate it is. However, we understand that, following the previous comment, sample size rather than income measurement is a major limitation in this case. Therefore, we have decided to drop this statement to make this section more straightforward/accurate. 

Thank you.

---

## [Decision Letter · Decision Letter 4]

17 Mar 2021

Obesity under Full Fresh Fruit and Vegetable Access Conditions

PONE-D-20-02163R4

Dear Dr. Silva,

We’re pleased to inform you that your manuscript has been judged scientifically suitable for publication and will be formally accepted for publication once it meets all outstanding technical requirements.

Kind regards,

Petri Böckerman

Academic Editor

PLOS ONE

Additional Editor Comments (optional):

Reviewers' comments:

Reviewer's Responses to Questions

**Comments to the Author**

1. If the authors have adequately addressed your comments raised in a previous round of review and you feel that this manuscript is now acceptable for publication, you may indicate that here to bypass the “Comments to the Author” section, enter your conflict of interest statement in the “Confidential to Editor” section, and submit your "Accept" recommendation.

Reviewer #2: All comments have been addressed

Reviewer #3: All comments have been addressed

2. Is the manuscript technically sound, and do the data support the conclusions?

Reviewer #2: Yes

Reviewer #3: Yes

3. Has the statistical analysis been performed appropriately and rigorously? 

Reviewer #2: Yes

Reviewer #3: Yes

4. Have the authors made all data underlying the findings in their manuscript fully available?

Reviewer #2: Yes

Reviewer #3: Yes

5. Is the manuscript presented in an intelligible fashion and written in standard English?

Reviewer #2: (No Response)

Reviewer #3: Yes

6. Review Comments to the Author

Reviewer #2: (No Response)

Reviewer #3: (No Response)

7. PLOS authors have the option to publish the peer review history of their article (what does this mean?). If published, this will include your full peer review and any attached files.

Reviewer #2: No

Reviewer #3: No

---

## [Editor Report · Acceptance letter]

8 Apr 2021

PONE-D-20-02163R4 

Obesity under Full Fresh Fruit and Vegetable Access Conditions 

Dear Dr. Silva:

I'm pleased to inform you that your manuscript has been deemed suitable for publication in PLOS ONE. Congratulations! Your manuscript is now with our production department. 

Kind regards, 

on behalf of

Professor Petri Böckerman 

Academic Editor

PLOS ONE